# CHALLENGING COMMON ASSUMPTIONS ABOUT CATASTROPHIC FORGETTING

## ABSTRACT

Standard gradient descent algorithms applied to sequences of tasks are known to induce catastrophic forgetting in deep neural networks. When trained on a new task, the model's parameters are updated in a way that degrades performance on past tasks. This article explores continual learning (CL) on long sequences of tasks sampled from a finite environment. **We show that in this setting, learning with stochastic gradient descent (SGD) results in knowledge retention and accumulation without specific memorization mechanisms.** This is in contrast to the current notion of forgetting from the CL literature, which shows that training on new task with such an approach results in forgetting previous tasks, especially in class-incremental settings. To study this phenomenon, we propose an experimental framework, *SCoLe* (Scaling Continual Learning), which allows to generate arbitrarily long task sequences. Our experiments show that the previous results obtained on relatively short task sequences may not reveal certain phenomena that emerge in longer ones.

## 1 INTRODUCTION

Continual learning (CL) aims to design algorithms that learn from non-stationary sequences of tasks. Classically, the main challenge of CL is catastrophic forgetting (CF) – fast performance degradation on previous tasks when learning from new data. CF is usually evaluated on scenarios with sequences of disjoint tasks (Lesort et al., 2020; Lange et al., 2019; Belouadah et al., 2021; Hadsell et al., 2020). In these scenarios, fine-tuning a model with plain empirical risk minimization objective and stochastic gradient descent (SGD) results in CF (Rebuffi et al., 2017; Lesort et al., 2019a; Kang et al., 2022).

The main motivation for this work is to step back from classical continual learning and investigate if fine-tuning with gradient descents approaches only truly leads to catastrophic forgetting or if it can result in knowledge accumulation and decaying forgetting. For example, Evron et al. (2022) theoretically showed that, for linear regression trained with SGD knowledge accumulation exists, leading to CF reducing uniformly when tasks reoccur randomly or cyclically. This indicates that CF might not be as catastrophic as it was initially assumed. Perhaps the problem is that CF is often studied in a setup where it is particularly excruciating — short task sequences with non-reoccurring data.

In this work, we empirically show that deep neural networks (DNNs) may consistently learn more than they forget when trained with SGD (Fig.1). We investigate how DNNs trained continually for single-head classification on long sequences of task with data reocurrence forget and accumulate knowledge.

To this end we propose *SCoLe* (Scaling Continual Learning), an evaluation framework for CL algorithms that generates task sequences of arbitrary length. As visualized in Fig. 2, *SCoLe* creates each new task from a randomly selected subset of all classes. A model is trained for some epochs on data from these classes until the task switches. The *SCoLe* framework creates tasks online

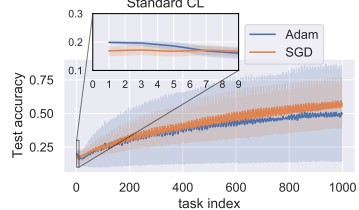

Figure 1: With SGD, knowledge accumulation is not observable in standard CL benchmarks (inset top). However, when repeating the sequence of tasks (bottom), knowledge accumulation is apparent and accuracy rises (MNIST, 2 per task, averaged over 3 lr and 5 seeds, each task trained until convergence).

by sampling an existing source dataset such as CIFAR (Krizhevsky et al.). By modifying the sampling strategy of the source dataset, we can control data recurrence frequencies, the complexity of the scenario, and the type of shifts in distribution (random, long-term, cyclic, etc.).

As in classical continual learning scenarios, *SCoLe* scenarios have regular distribution shifts that should lead to CF. However, in *SCoLe*, tasks and data can sparsely reoccur. This reoccurrence makes it possible when training with naive SGD, to study if knowledge can accumulate through time. The idea is: if the model does not forget catastrophically but progressively, some knowledge can be retained until the next occurrence of the data leading to potential knowledge accumulation. We study the impact of such reoccurrence in a series of *SCoLe* experiments and confirm that DNNs trained with gradient descent only accumulate knowledge without any supplementary CL mechanism.

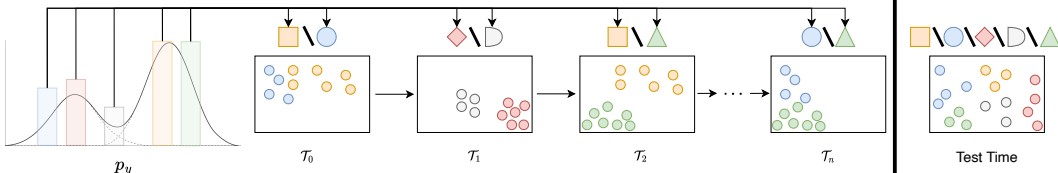

Figure 2: Illustration of *SCoLe* scenario. With 5 classes in total (one per color) and 2 classes per task. The data are selected randomly based on their label to build the scenario dynamically, into a potential infinite sequence. The evaluation is performed on the test set containing all possible classes.

Our contributions are as follows: (1) We propose an experimentation framework "*SCoLe*" with a potentially infinitely long sequence of tasks. *SCoLe* scenarios are built to study knowledge retention and accumulation in DNNs with non-stationnary training distributions. (2) We show that in such scenarios standard SGD retains and accumulates knowledge without any CL algorithm, i.e. without supplementary memorization mechanism. This result is counterintuitive given the well-known phenomenon of catastrophic forgetting in DNNs. (3) We study the capabilities and limitations of such training with scenario based on a variety of datasets (MNIST, Fashion-MNIST, KMNIST, CIFAR10, CIFAR100, miniImageNet) and scenarios.

## 2   *SCoLe*:A FRAMEWORK FOR CL WITH LONG TASK SEQUENCES

**General Idea.** We propose a framework that allows the creation of an arbitrarily long sequence of tasks. The setting is based on the finite-world assumption (Mundt et al., 2020), which hypothesizes that the world has a finite set of states, and in a finite period of time, the agent will see all of them. Later data will, therefore, necessarily be a repetition of previous ones. As in classical CL, in *SCoLe* each task is comprised of a subset of world's data, whereby the evaluation is done on the whole world. In such a world, a learning system must accumulate knowledge about the world by experiencing parts of it in isolation. The difference to the classical CL scenarios is that data sparsely reoccurs. An agent can succeed in this setup only by accumulating knowledge faster than forgetting it. *SCoLe* can reveal learning dynamics of DNNs under distribution shift that are not observable on short sequences of non-overlapping tasks, as we show next. Measuring the performance on whole world allows us to estimate whether, overall, the agent accumulates knowledge faster than it forgets.

**Framework.**

We instantiate this idea in a classification setting (Fig. 2). At each task, a subset of classes is randomly selected from the total of $N$ available classes ($N$ is dataset dependent). The agent is a DNN that learns to classify on this subset only and is tested on the full test set with all classes. The framework considers scenarios with varying numbers of tasks $T$ and classes per task $C$.

Formally, the training set $D_t$ for a task $t$ consists of $(x, y)$ pairs sampled from the distribution $p(X, Y|S_t) = p(X|Y)p(Y|S_t)$. Here $S_t = \{c_i\}_{i=0}^{C-1}$ is the set of classes in task $t$. In the default *SCoLe* scenario, the elements of $S_t$ are sampled from the uniform distribution over all $N$ classes $c_i \sim U(0, N-1)$ without replacement. We also consider cases where $p(S_t; C)$ is non-uniform (Sec. 4.2) or evolves over time (Sec. 5). Label $y$ is sampled uniformly over $S_t$ and $x$ is obtained as $x \sim p(X|Y = y)$. The test set $D_{test}$ contains all classes in the scenario. Following the data generation process, it is generated with $C = N$.

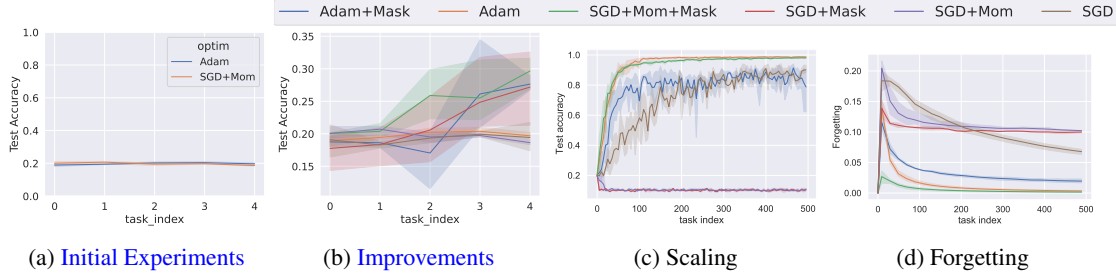

(a) Initial Experiments    (b) Improvements    (c) Scaling    (d) Forgetting

Figure 3: Initial experiments with default hyper-parameters on MNIST. Masking gradient and removing momentum, increase knowledge accumulation leading to a reduction of forgetting.

**Measuring divergence to IID.** *SCoLe* enables studying knowledge accumulation as a function of divergence to IID of the training distribution. We formalize the closeness of *SCoLe* scenarios to the identically and independently distribution (IID) training regime using the expected (over tasks) KL-divergence. When $p(Y)$ is uniform over $N$ classes, the KL-divergence between $p(X, Y|S_t)$ and the IID joint $p_{iid}(X, Y)$ is $\mathbb{E}_t[D_{KL}\big(p_t(X, Y|S_t) \mid\mid p_{iid}(X, Y)\big)] = \log \frac{N}{C}$ with $N$ the total number of classes and $C$ the number of classes per task. (for derivation cf. App. B). Hence, in *SCoLe* scenarios using an $N/C$ ratio closer to 1 generates training tasks closer to IID. It also reduces the expected number of tasks between two occurrences of the same task $\tau_{task}$ (or of a class $\tau_{class}$). Finally, it increases the probability $p_o$ that classes of consecutive tasks overlap (cf. App. A for details). For example with $C = 2$, $\tau_{class} = 45$ and $\tau_{task} = 5$ for CIFAR10, while $\tau_{class} = 4095$ and $\tau_{task} = 50$ for CIFAR100. Thus, revisiting exactly the same task in CIFAR100 for $C = 2$ is very rare (every 4,095 tasks). Note that in standard CL $\tau_{task}$ and $\tau_{class}$ are inf while $p_o$ is 0.

In this paper, we propose three sets of experiments. We first present how to observe knowledge accumulation (Sec. 3). Secondly, we experiment with various levels of non-stationarity of the scenarios to understand better the robustness of knowledge accumulation with gradient descent (Sec. 4), finally we observe knowledge accumulation with long term distribution shifts (Sec. 5).

## 3 OBSERVING KNOWLEDGE ACCUMULATION IN *SCoLe*

In this section we show how, in *SCoLe*, we can learn a sequence of tasks without a continual learning algorithm such as replay, regularization, or a dynamic architecture.

### 3.1 INITIAL EXPERIMENT

In this experiments, we estimate knowledge accumulation with popular optimizers.

**Setting:** We study several standard approaches for training DNNs. We use the two most popular optimizers in CL and ML in general: SGD with momentum (Qian, 1999), and Adam (Kingma & Ba, 2014). For SGD, we set the momentum to 0.9, since it is the default value in PyTorch (Paszke et al., 2019). We start to experiment with a scenario with only 20 tasks on the MNIST dataset with 2 classes per task and a small convolutional neural network (c.f. Appendix I).

At task $t$, the objective is to find $\theta_t^*$ that minimizes the (global) test set $D_{test}$ loss. This is done through the proxy of training loss minimization (cross-entropy) $\ell_{(x,y)\in D_t}(f(x; \theta), y)$, where $f(\cdot; \theta)$ is a function realized by a neural network parameterized with $\theta$. Training is done using batched training procedure with $b$ denoting the batch size and $e$ denoting the number of epochs per task. We define $u_t$ to be the number of gradient updates per task, which in general case is $u_t = \left\lceil \frac{|D_t|}{b} \right\rceil * e$.

**Results:** The results in Fig. 3a show no progress in the test accuracy, even with a decrease in performance over time. This result is in accordance to the common CF phenomenon (French, 1999).

Next we are going to show that knowledge can actually accumulate as seen in Figure 1 and how to facilitate such accumulation with minimal modifications to the learning algorithm.

## 3.2 Improving Knowledge Accumulation

**Setting:** We propose to use two small improvements for gradient descent (1) removing the momentum for SGD, (2) masking gradient for classes not currently in the task in the output layer: "gradient masking". Accumulation can be observed without those improvements as seen in Fig. 1 but with those improvements it becomes easier and more consistent. We test those improvements on the 20 tasks scenarios and on a 500 tasks scenarios ( 2 classes per task).

Intuitively, in the presence of data distribution drifts, momentum (also present in Adam) produces a mixture between the gradient of the previous task and the gradient of the current task. This can create interference in the training process.

Recent papers show that masking the gradient in the last layer helps classifiers to learn continually (Caccia et al., 2022; Zeno et al., 2018) even without any supplementary memorization process (Lesort et al., 2021a). Therefore, we tested whether "group masking" from Lesort et al. (2021a) could also be useful in training a model end-to-end in a *SCoLe* scenario. The idea is that, when learning on subsets of classes (without replay), the gradient is only backpropageted through the outputs corresponding to the classes in the current batch. The inference is realized in the same way as single head approaches but the loss is computed by only taking into account outputs of classes currently in the mini-batch. On the other hand, Ramasesh et al. (2021) shows that CF happens the most on the higher layers and gradient masking is the most straight-forward way to reduce forgetting in the output layer for free. We hypothesize that apparent reduction of forgetting through such masking can be partially attributed to its regularizing effect for preventing overfitting and not forgetting (even though in this case both are evidenced in the reduced performance): if no masking is applied the layer can quickly achieve low training error by reducing biases and norms of classes not observed in the current batch hence Hou et al. (2019); Lesort et al. (2021a).

**Results:** Fig. 3b shows that these two small improvements result in an increase in performance even after a few tasks. In Fig. 3c, we scale the number of tasks to see at which point the accuracy stops improving. Thereby, the accuracy of the model can reach close to IID a accuracy.

Those slight improvements help us see that gradient descent algorithms may not forget catastrophically even without using memorization mechanisms (beyond masking in the last layer).

## 3.3 HPs search.

**Setting:** To find the best setup to run our further experiments, we run a small hyperparameter search on the same scenario, but with 500 tasks on MNIST, Fashion-MNIST, and KMNIST. Fig. 4a shows the average performance on the three datasets with various learning rates.

**Results:** In Fig. 4b we show that SGD without momentum and with masking is a stable baseline, as it leads to knowledge accumulation consistently on all datasets. We also find that, with a small learning rate, Adam and SGD with momentum can achieve knowledge accumulation and improve their accuracy on the test set when scaling the number of tasks, even without masking.

Our results show that knowledge accumulation might happen with any gradient descent optimizer. SGD with masking (without momentum) is overall more stable. Hence, We will experiment this baseline in further experiments.

## 3.4 Robustness of Knowledge Accumulation

We investigate if knowledge accumulation happens consistently among datasets and architectures.

**Setting:** We create scenarios on MNIST, Fashion-MNIST and KMNIST with 500 tasks, CIFAR10, CIFAR100 and miniImageNet with 1,000 tasks (3 seeds). We experiments the baseline SGD+Mask on all of them. For CIFAR100 and miniImageNet, we set $C = 5$, while for the other datasets $C = 1$. In addition, we train several architectures (Resnet18, Inception, vit_b_16 and VGG) from the torch library and compare them in a default *SCoLe* scenario on CIFAR10 with 2 classes per task.

**Results:** Fig. 4b shows the learning curve on the various datasets. We normalize accuracies by the IID accuracy to make curves comparable. The IID test accuracies are: MNIST $99\%$, Fashion MNIST $89\%$, KMNIST $94\%$, CIFAR10 $79\%$, CIFAR100 $40\%$, and miniImageNet $20\%$. The accuracies

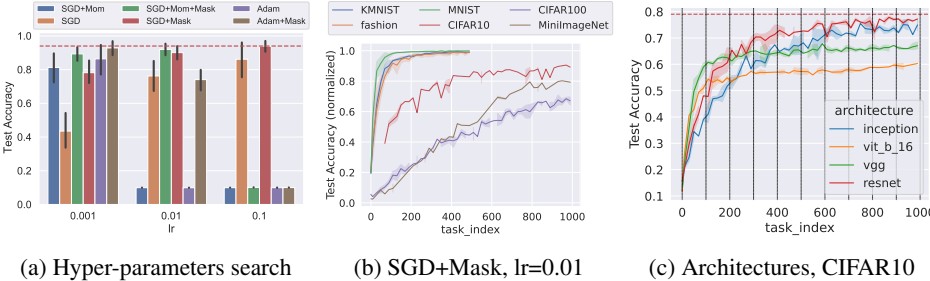

(a) Hyper-parameters search     (b) SGD+Mask, lr=0.01     (c) Architectures, CIFAR10

Figure 4: Test acc. with $T$ scaled up to 1,000 tasks averaged over MNIST, Fashion-MNIST, KMNIST with 3 seeds. - - - line is the best IID performance. SGD+Mask w/tMomentum is the most stable baseline that works with various datasets and architectures.

were obtained with the same models and Adam training with default parameters and without data augmentation. This figure shows that knowledge accumulation occurs consistently in all these datasets. Fig. 4c shows that on CIFAR10 the knowledge accumulation consistently happens with various type of architectures.

**Conclusion.** In this section, we showed that SGD without momentum and with gradient masking accumulates knowledge over time. After a high number of tasks, it can even converge to a solution that is close to the IID training accuracy. This shows the limited effect of CF and the ability of models to retain and accumulate knowledge.

A possible explanation for this behavior is that, while learning new tasks, DNNs do not forget catastrophically. Beyond the catastrophic drop in performance on previous tasks, some knowledge stays and is forgotten only progressively. If data reoccurs, as in *SCoLe*, before being completely forgotten, knowledge accumulation happens and leads to a progressive diminution of forgetting. Hence, the model learns more than it forgets, and the accumulation of knowledge overcomes the forgetting. In CL without masking, forgetting occurs more strongly because the last layer is determinant for prediction and more sensitive to forgetting (Wu et al., 2019; Hou et al., 2019; Zhao et al., 2020; Ramasesh et al., 2021; Bell & Lawrence, 2021; Lesort et al., 2021a). Masking stabilizes the last-layer weights, and when we scale the number of tasks with reoccurring data, knowledge retention and accumulation is amplified and becomes visible.

## 4 EFFECT OF NON-STATIONARITY ON CONTINUAL LEARNING WITH SGD

In this section, we turn out attention to various levels of non-stationarity in *SCoLe* scenarios. Specifically, we study three ways to control the level of non-stationarity: (1) through the divergence from the IID by changing the total number of classes $N$ and classes per task $C$, (2) by changing the probability of classes reoccurrence, and, (3) through the frequency of the distribution shift by changing the number of gradient steps per task $u$.

### 4.1 INCREASING THE DIVERGENCE TO IID BY INCREASING THE NUMBER OF CLASSES

Here we control the divergence to IID by varying the total number of classes $N$ in *SCoLe* scenarios with fixed $C = 2$.

**Setting:** We use CIFAR100 and create different subsets of $N$ classes by subsampling the total number of classes of the train **and** the test set. A higher $N$ results in a more difficult learning problem (more classes in the test set) and a greater divergence from the IID regime (i.e. larger $\tau_{task}$ and $\tau_{class}$). We fix number of epochs $e = 1$ and batch size $b = 64$ resulting in $u = 15$ ($|D_t| = 900$). **Results:** In Fig. 5 (left) we plot the accuracy of scenarios with different $N$. We normalized the test accuracy by the IID accuracy on the same data in order to only assess the effect of increasing $N$ on the knowledge accumulation and minimize the effect of increasing the difficulty of the problem.

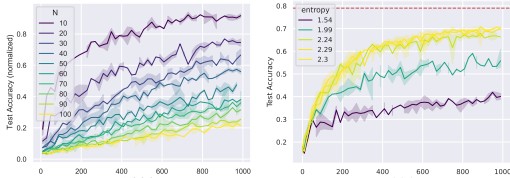 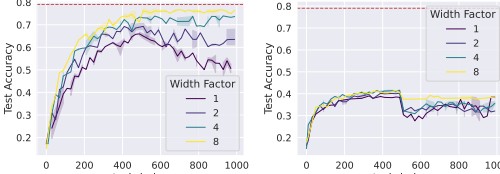

Figure 5: Varying $N$ on CIFAR100 dataset (left) for C=2,$N$ impacts divergence from IID and difficulty.(Right) Imbalanced class distribution on CIFAR10. - - - line is the IID training baseline.

Figure 6: (Left) Test acc. when half of the classes is removed at T=500. (Right) Test acc. under a shift in classes distribution at T=500. The impact of shift is smaller for wider models.

This experiment shows that widening the divergence from IID, slows down knowledge accumulation but it still occurs. Reaching IID accuracy would presumably be possible but would require further scaling $T$ and/or $C$ (cf App. Fig. 10).

## 4.2 CHANGING CLASS OCCURRENCE FREQUENCY

While so far, the task sampling distribution $p(Y)$ was uniform, we now make it non-uniform by changing its entropy. Hence, $\tau_{class}$ will grow for some classes and decrease for others $N = 10$ is fixed.

**Setup.** We plot the model's test accuracy curves for different entropy of $p(Y)$. As in previous experiments, the class distribution does not change over time in a given scenario and instances per classes are balanced within a task. Details on the implementation of the entropy control are in App. G.2. $Entropy = 2.3$ corresponds to the uniform distribution. **Results:** In Fig. 5 (right), we see that the closer we are to a uniform class distribution for sampling tasks, the steeper is the accumulation curve. This can be attributed to nonuniform $\tau_{class}$ over classes, leading to (a) knowledge accumulation becoming slower for these rare classes and (b) more probable classes are repeated more frequently in the same context, leading to possible overfitting.

Lowering the entropy of the class distribution increase the sparsity of reoccurrence of certain classes, which leads to a global slowdown of knowledge accumulation.

## 4.3 THE ROLE OF THE NUMBER OF UPDATES BETWEEN REVISITS

In previous experiments, we have shown that knowledge accumulation reliably occurs for one epoch per task ("online", $e = 1$). Here we study the effect increasing the number of epochs per task ("offline"). Larger $e$ should result in larger expected number of SGD steps between re-occurrence of classes, $\overline{u}_{class}$, and more time for CF.

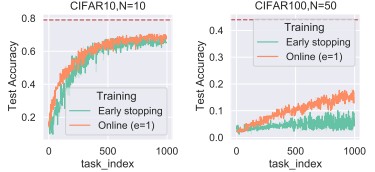

**Setup.** We compare the online setup ($e = 1$) with the offline one. In the latter, we train on each task until convergence with an early stopping (e.s.) criterion of 5 epochs (e.s. is the strategy of choice for model selection). We fix the batch size $b = 64$, classes per task $C = 2$ and study CIFAR10 ($N = 10$) and CIFAR100 datasets ($N = 50$). CIFAR10 has 4500 samples/class resulting in $u = 141$, and $\overline{u}_{class} = u * \tau_{class} = $

Figure 7: (left:) Test acc. in the online setting ($e = 1$) versus the offline setting (train until convergence) with $N = 10$. (right:) Same as left, but with $N = 50$.

705 in the online case. In the offline setup, $\tau_{class}$ and $\overline{u}_{class}$ are at least 5 times bigger due to e.s. criterion. On the other hand, CIFAR100 dataset comes with 450 samples/class resulting in $u = 15$ but comparable $\overline{u}_{class} = 15 * 50 = 750$ due to higher $\tau_{class}$ (cf.Sec. 2).

**Result.** In Fig. 7(left) for CIFAR10 the gap between the online and offline training is small showing that increasing the number of gradient updates does not necessarily slow down knowledge accumulation. On this dataset knowledge accumulation dominates forgetting. For CIFAR100 (right) the gap is significantly larger. Given the comparable period size $\overline{u}_{class}$, this can be explained by (a) a lack of data diversity given the smaller number of samples per class which can lead to overfitting, and (b) a

larger $\tau_{class}$. The latter implies that a (much) larger set of tasks with different classes is seen before an instance of a class is repeated, which can lead to more forgetting. Hence, the number of samples per classes and $\tau_{class}$ are critical for accumulation.

Increasing the number of gradient steps per task can compromise knowledge accumulation when there is a risk of overfitting. Nevertheless, it can also keep it high by improving the knowledge of the model on the current tasks. A right trade-off between learning new knowledge and forgetting past ones exists then that does not necessarily push to low divergence with the IID setting. We propose other evidence of this, by varying the number of epochs per task and the batch size in appendix K Fig 17.

**Conclusion.** The *SCoLe* framework is convenient for controling the stationarity of the scenarios or creating setups of various difficulty to study knowledge accumulation. In this section, we have shown that knowledge retention and accumulation still happens when increasing the non-stationary. This means that knowledge learned in the past can be retained for long periods of training without being forgotten. In the next section, we evaluate how knowledge retention and forgetting happen in *SCoLe* scenarios with long-term distribution shifts that influence many tasks.

## 5 EFFECT OF LONG-TERM DISTRIBUTION SHIFTS

In the experiments presented so far, we sampled classes for each task from the same distribution $p(S_t; C)$. Here we create *SCoLe* scenarios with a shift over time in the class distribution and assess the accumulation of knowledge under long-term shifts — shifts that persist for over several hundreds of tasks. Those shifts may result in very large or infinite $\tau_{class}$ for some classes. We evaluate knowledge retention and interference by designing three different shift patterns and evaluate models with increasing width to access whether increasing the width can slow down forgetting (Mirzadeh et al., 2021).

### 5.1 KNOWLEDGE RETENTION WITH LONG-TERM CLASS SHIFT

We assess the capabilities of deep neural networks to maintain correct prediction on classes that stop appearing. This evaluation of knowledge retention is strict in comparison to related works (Fini et al., 2022; Davari et al., 2022) that evaluate knowledge retention by linear probing latent representation. In our setup, knowledge retention in lower layers is not sufficient, and the model also has to maintain knowledge from feature extraction in lower layers to the decision boundaries in the last layer.

**Setting:** We train on a scenario with $C = 2$ and uniform $p(Y|S_t)$ on CIFAR10. We start the training with all classes, after 500 tasks, we remove half the classes from the class distribution. In contrast to standard CL, where distribution shift is usually caused by adding classes, here we remove classes. In such a scenario, the forgetting behavior should be smooth because the error on remaining classes should be low. **Results:** Fig. 6 (left) shows that even if no new data is introduced to the learner, that is, no interference with old knowledge is possible, the model can still forget when a subset of already learned classes is no longer trained on. Interestingly, in this setup forgetting is slow and not catastrophic, and knowledge persists during several hundreds of tasks. Moreover, we clearly observe that growing the width of the model increases knowledge retention to the point that it looks like the model perfectly memorized removed classes for the maximum model size.

### 5.2 KNOWLEDGE RETENTION WITH CLASS SUBSTITUTION

Similarly to the previous section, we investigate a setting with an abrupt shift in the class distribution, however instead of removing classes (shrinking the domain of $p(S_t; C)$), we replace existing classes with new (shift the domain of $p(S_t; C)$). This allows us to investigate the interference and forgetting dynamics that both cause performance drop in this setting. The goal is to assess whether observed knowledge retention can help to slow down forgetting.

**Setting:** First, 500 tasks are generated from the first 5 classes of CIFAR10 (first period), and the second 500 tasks are generated from the remaining 5 classes (second period). There is then no overlap between the classes in the first and second periods. Also here we test models with various widths. **Results:** The results in Fig. 6 (right) show that this sudden class shift creates disturbance in the training process as in classical CL scenarios. Moreover, in the second period of training, the model

struggles more to learn the tasks than during the first period, meaning the first period does not provide good initialization for later tasks and that the forward transfer is limited in such a regime. This result corroborates the results of Ash & Adams (2020), who witnessed a similar phenomenon in a transfer setting. Similarly to the previous section, observe that wider models can better resist to CF. Still, this difference appears less clear, and it could be due to the better knowledge retention of first classes.

## 5.3 CYCLIC SHIFTS

Here we aim to assess whether knowledge retention in DNNs observed previously persists when we fix $\tau_{class}$ to be equal for all classes but varying number of expected SGD updates $\overline{u}_{class}$ before repetitions. To this end we let the distribution $p(S_t; C)$ follow a cyclic patterns.

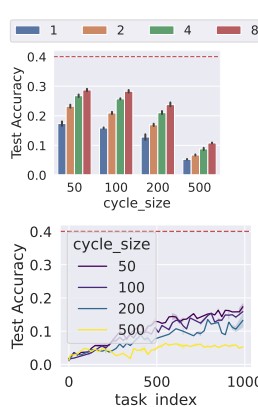

**Setting:** We define a shifting window of $W$ classes as the domain of $p(S_t; C)$ for $\frac{N}{\lambda}$ tasks after which it is shifted by one class or more depending on $N$ and $\lambda$. Here $\lambda$ is the cycle size, i.e. $W$ is exactly the same every $\lambda$ tasks. For example, if the class subset $W_t$ before is $[0, 1, 2]$, after a shift it will become $[1, 2, 3]$. After $\lambda$ tasks, it will be again $[0, 1, 2]$. We use the full CIFAR100 dataset ($N = 100$) with 5 classes per task, the window size is $W = 10$, and the cycle size is $\lambda \in [50, 100, 200, 500]$, higher $\lambda$ leads to higher $\overline{u}_{class}$. We choose CIFAR100 because it has more classes than with CIFAR10 and allows creating of shifts with longer period. In this experiment, when a class leaves the subset window, it needs $\lambda - W$ tasks to return leading to equal $\tau_{class}$ for all classes.

**Results:** We can see in Fig. 8 that increased shift period makes training harder, i.e., better learning happens if classes reoccur more frequently. Even with the largest period, the model still progresses systematically over time, as seen in Fig. 8 (bottom). Consistently with other experiments, wider models result in better performance, as shown in Figs. 8 (top). These experiments show that SGD-trained DNNs are also capable of long-term retention and can still accumulate even if they do not see some classes for a long period of time.

Figure 8: Cyclic Shifts Experiments: (top) shows the test accuracy averaged over the 10 last tasks for all cycle and model sizes. (bottom) the effect of cyclic shifts for the smaller model.

**Conclusion.** Forgetting still happening in long sequences of tasks with long term distribution shifts is expected. However, experiments in this section show that models are capable of long-term knowledge retention, enabling knowledge accumulation even when data is not seen for a long time. Our results are in line with the findings of Mirzadeh et al. (2021): Wider models forget less in incremental scenarios. Further, we have also shown that the widest models are capable of almost perfect knowledge retention and that with long-term distribution shifts such as cyclic shifts, deep neural networks can accumulate knowledge. They can memorize and reuse knowledge from classes not seen since more than 200 hundred tasks.

## 6 RELATED WORK

Most scenarios in the continual learning literature study catastrophic forgetting (van de Ven & Tolias, 2019; Lesort et al., 2021b). They are made up of a sequence of tasks where data points appear in one task without reoccurring later. Those settings evaluate whether models can remember tasks that they have seen only once. The no reappearance constraint makes the evaluation of CF clearer. Tasks overlap can not interfere with forgetting. However, they cannot evaluate if models accumulate knowledge through time that could be reused with data/tasks reoccurence.

The scenario we propose has some similarities to the CMR (Lin et al., 2022) and OSAKA (Caccia et al., 2020) frameworks. However, in our setting, we do not evaluate fast adaptation, but rather the capacity to learn a solution to a problem from a long sequence of subproblems. Moreover, there is no real concept shift in *SCoLe*, that is, $p(y|x)$ is fixed over time. Our evaluation protocol is also similar to the ALMA scenario (Caccia et al., 2021), since we evaluate on a fixed test set. However, in ALMA the data distribution does not drift, while in *SCoLe* there are drifts between tasks. A scenario with a long sequence of tasks was proposed by (Wortsman et al., 2020). Their scenario is composed of various permutations for the permut-MNIST scenario. This scenario also makes it possible to scale

the number of tasks, but it does not allow one to see the progressive knowledge accumulation of SGD witnessed in *SCoLe* scenarios because there is no task reoccurence.

In Lesort et al. (2021a), the masking method, referred to as "group masking", applies gradients only to the outputs of classes within the current batch while training. In particular, the authors showed that this masking is possible while training a linear classifier on top of a frozen pretrained model. Caccia et al. (2022); Zeno et al. (2018) applied a similar method in continual learning, respectively, in end-to-end training and multihead training, and showed that this method helped mitigate forgetting.

## 7 Discussion

Another interpretation of knowledge accumulation (besides the forgetting lens) is through the lens of transfer. Knowledge accumulation is also the result of a positive forward transfer (with SGD). The knowledge learned on a task can be reused and improved when the task reoccurs. By this perspective, avoiding forgetting and maximizing forward transfer to the next occurrence, both require some form of knowledge accumulation through time.

Beyond evaluating how SGD forgets, *SCoLe* is also designed to be a practical evaluation tool to assess knowledge accumulation in a setting with data reoccurrence. *SCoLe* could be directly transferable to similar real-life settings. For example, a robot that manipulates objects from visual perception or performs semantic segmentation of scenes when moving around. The robot can witness data distribution shifts through time depending on different factors (lightning, rooms, objects), and the same tasks/classes naturally reoccur. Our results suggest that SGD with gradient masking could be a very light approach that could serve an embodied agent with restricted compute or memory.

The reoccurence of data could be compared with the action of a replay methods. However, while replay typically emulates an IID distribution, this never happens in *SCoLe* scenarios. The data streamed by the generative process only provides access to a subset of the classes. Moreover, in replay-based methods, the amount of replay and the distribution of replay can be adapted to maximize performance while *SCoLe*'s generative process can not be modified during training.

Training dynamics with *SCoLe* also share similarities with reinforcement learning agents that learn in finite environments, such as video games or simulations. While they learn, the training distribution changes, either because the policy changes or because the agent explores new parts of the environment. This leads to drifts in the local data distribution with reoccurring data as in the *SCoLe* scenarios. In the long run, reinforcement learning agents can still learn a policy that is applicable to the entire environment. The results of this paper are in line with this behavior.

## 8 Conclusion

*SCoLe* is a framework for creating continual learning scenarios with long sequences of tasks. It can be adjusted through using several hyper parameters to generate scenarios of varying difficulty and various distribution shifts between tasks. This versatility makes *SCoLe* a tool to study the ability of deep neural networks to continually learn.

This paper shows that beyond the catastrophic decrease in performance in classical continual learning benchmarks, standard gradient descent is capable of knowledge retention and accumulation. The phenomenon becomes visible when we scale the number of tasks and repeat classes and tasks. To our knowledge, this phenomenon has not been investigated in the literature before. To amplify knowledge accumulation and to increase its impact, we propose to mask gradients of unused classes in SGD training and removing momentum. In the classical continual learning literature the common intuition is that such a simple approach would not be enough to allow accumulation and overcome *catastrophic forgetting*. However, our results show that knowledge retention and accumulation are sufficient to overcome catastrophic forgetting in *SCoLe* scenarios. The magnitude of this behavior still depends on the scenario characteristics and notably on the frequency of reoccurence. Our experiments with *SCoLe* show interesting insights into forgetting, knowledge accumulation, and knowledge retention. We believe that the knowledge accumulation observed with SGD is not restricted to *SCoLe* training scenarios, the training setups was made to amplify knowledge accumulation. We hope that our framework will inspire further research in this field improve knowledge accumulation in deep neural networks in various scenarios.

## REPRODUCIBILITY STATEMENT

For all of our experiments, we use the standard deep neural network architectures and publicly available datasets. We specify all the training details, including datasets for the experiments of knowledge accumulation and distribution shifts in section 3 and 4 respectively. We also show the *SCoLe* scenario in Fig. 2 and include the pseudo-code for implementation in Appendix D. We provide the code (https://anonymous.4open.science/r/COVERGENCE-5533/) for all the experimental results presented in the paper. The code base contains all the parameter configurations for running the experiments to reproduce the figures.

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

## A    REOCCURRENCE STATISTICS

The re-occurrence statistics for *SCoLe* include (assuming the uniform distribution of classes):

- the expected number of tasks between the same class is repeated $\tau_{class} = [\binom{N-1}{C-1}/\binom{N}{C}]^{-1}$

- the expected number of tasks between the task is repeated $\tau_{task} = \frac{1}{p}$, where $p$ is the probability of sampling each task, $p = 1/\binom{N}{C}$.

- $p_o$ – the probability that the supports $S_t$ and $S_k$ for two tasks ($t \neq k$) overlap is $\sum_{i=1}^{C}[\binom{C}{i}\binom{N-C}{C-i}]/\binom{N}{C}$.

Note that in standard CL $\tau_{task}$ and $\tau_{class}$ are $\infty$ while $p_o$ is 0. In vanilla experience replay (ER) (Rolnick et al., 2019; Rebuffi et al., 2017) – which is a method for mitigating CF (cf Sec.6), both $\tau_{class}$ and $\tau_{task}$[1] are 0, and $p_o = 1$ [2].

---

[1] Since ER simulates IID setup, all tasks overlap and technically speaking no tasks are identical in this regime.
[2] In "standard" CL with ER this in only true for future tasks w.r.t past tasks.

# B  KL DERIVATION

$$D_{KL}\big(p(X,Y|S_t)||p_{iid}(X,Y)\big) = \sum_{x \in X, y \in Y} p_t(x,y|S_t) \log \frac{p_t(x,y|S_t)}{p(x,y)} =$$

$$= \sum_{x \in X, y \in Y} p(x|y)p(y|S_t) \log \frac{p(x|y)p(y|S_t)}{p(x|y)p(y)} = \sum_{y \in Y} p(y|S_t) \log \frac{p(y|S_t)}{p(y)} \sum_{x \in X} p(x|y) =$$

$$= \sum_{y \in Y} \frac{\mathbb{1}_{\{y \in S_t\}}}{C} \log \frac{N}{C} = \log \frac{N}{C}$$

Since $p(y|S_t)$ is uniform over elements n set $S_t$, it is equal to $\frac{1}{C}$ and the uniform $p(Y)$ is evaluated to $\frac{1}{N}$. Since each of the $T$ tasks is equally probable the expected divergence is as follows:

$$\mathbb{E}[\log \frac{N}{C}] = \sum_{t=1}^{T} \frac{1}{T} \log \frac{N}{C} = \log \frac{N}{C}.$$

# C  LIMITATIONS

In this paper, we investigate the phenomenon of steady knowledge accumulation until convergence when continually training a model in long sequences of tasks within a finite world despite catastrophic forgetting. However, with increasing complexity of tasks, it takes longer to converge due to slower knowledge accumulation.

Estimating the complexity of a *SCoLe* scenario is challenging and probably impossible in practice. In this condition, it is not clear how long the sequence of tasks needs to be for a given scenario.

It is not guaranteed that the trends observed in this work could be observed in the same way for more challenging datasets such as the ImageNet (Deng et al., 2009). However, the size of the chosen datasets make *SCoLe* ideal for fast experimentation, which has helped us to test a large number of hypotheses. This is an advantage that would not exist with larger datasets.

# D  SCENARIO IMPLEMENTATION

The implementation presented in Figure 9 proposes a static version of the scenario. However, the "probability" distribution can be modified through the task sequence to create never-ending drifts or cyclic drifts in the class distribution or simply to change the balance of the class distribution.

# E  TRAINING IID BASELINES

The training of the IID baselines has been carried out with the same models as the other training processes. However, they were trained with Adam optimizer to improve performance. They were trained with 100 epochs on the full dataset. We experimented with the learning rates of $0.001$, $0.01$, and $0.1$ over 5 seeds and only kept the best-performing baselines.

# F  ADDITIONAL EXPERIMENTS

## F.1  INCREASING THE NUMBER OF CLASSES PER TASK

Fig. 10, shows that our training framework also works when the number of classes per task increases. The more classes in the task, the faster the learning curve is. Here, we sample tasks from the entire CIFAR100 dataset.

## F.2  MAKING THE MODEL DEEPER

In these experiments, we train on CIFAR10 with binary tasks and classes sampled uniformly. Fig. 11 shows that depth has a low impact in our experiments.

```python
# num_classes: the total number of classes
# classes_per_tasks: number of tasks per class (2 by default)
# probability: vector defining probability of sampling each class
#                                   for a task (Uniform by
#                                   default)
# nb_epochs: epochs of training per task (1 by default)

import numpy as np
from continuum.scenarios import ClassIncremental
from continuum.datasets import CIFAR10

scenario = ClassIncremental(CIFAR10(config.data_dir, train=True),
                                   nb_tasks=nb_classes)
test_set = CIFAR10(config.data_dir, train=False).to_taskset()

for task_index in range(num_tasks):
    classes = np.random.choice(np.arange(num_classes), p=
                                          probability, size=
                                          classes_per_tasks,
                                          replace=False)

    # create taskset with only selected classes
    taskset = scenario[classes]
    for epoch in range(nb_epochs):
        # train the model on "taskset" data
        [...]
    # test the model on the full test set
    [...]
```

Figure 9: Pseudo-Code using continuum (Douillard & Lesort, 2021) to control the distribution imbalance in classes.

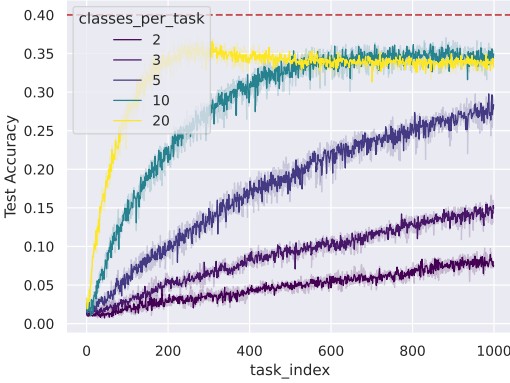

Figure 10: Growing the number of classes per task within a task in full CIFAR100 dataset.

### F.3 STRUCTURING THE TASK SEQUENCE

**Setting:** In this experiment, we want to evaluate the role of randomization of classes within tasks. In other words, we try to answer the following question: Is it important that classes are randomly sampled when building tasks? For this, we start with a fixed sequence of binary classification tasks. This sequence is built so that all possible pairs of classes exist and occur only once. By default, training is achieved by repeating training on this fixed sequence of tasks until the end of the full sequence of tasks. We compare this baseline with the same sequence of tasks, but for each task, we set the probability $p$ that each class is flipped by random to another class. The classes of the initial sequence of tasks is $[0, 1] \rightarrow [0, 2] \rightarrow [...] \rightarrow [1, 2] \rightarrow [1, 3] \rightarrow [8, 9] \rightarrow [0, 1] \rightarrow [...]$. The values of $p$ are $0, 0.01, 0.1, 0.2, 0.3, 0.4, 0.5, 1$. We can see that $p = 1$ is the same as in the default scenario.

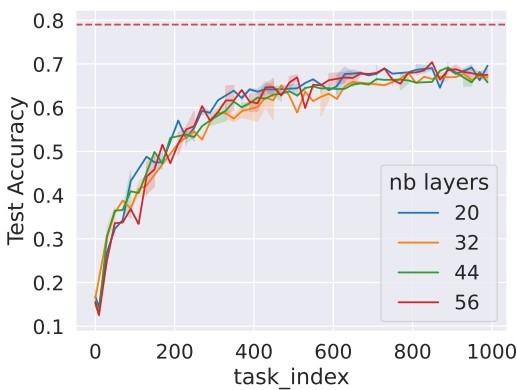

Figure 11: Growing the number of layers in the resnet model. - - - - - line represent IID performance with resnet22.

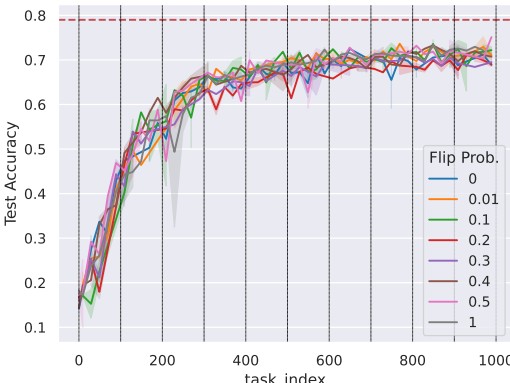

Figure 12: Comparison on several sequence of tasks with a default structure modified by random flip of classes at each task. Scenario created with CIFAR10, 2 classes per task. The randomization of tasks is not critical for knowledge accumulation.

**Results:** The results presented in Fig. 12 show that having a fixed sequence of tasks instead of a randomized one does not reduce knowledge accumulation in our setting.

### F.4 LIMITING THE POSSIBLE PAIRS OF CLASSES IN TASKS

**Setting:** In this experiment, we want to evaluate how important it is that all possible pairs of classes exist within the full sequence of tasks. For this, we start from the list of all possible tasks and select only a subset of them. When building the task sequence, we only select pairs of classes from the selected list.

**Results:** The results presented in Fig. 13 show that the presence of all possible pairs of classes within the sequence of tasks plays an important role. In fact, without replay to learn discriminative features between two classes, classes must be on the same task (Lesort et al., 2019b).

## G INCREASING DIFFICULTY

### G.1 REGRESSSIONS

### G.2 MODIFICATION OF DISTRIBUTION ENTROPY

To change the entropy of the class distribution, we start with a uniform vector of probabilities $u$. For each class, $u$ gives the probability of this class to be sampled for a task. To create an imbalance

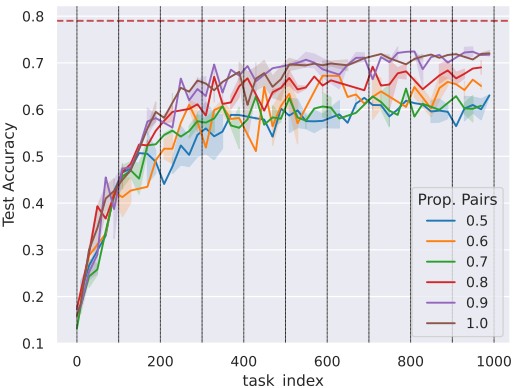

Figure 13: Comparison of *SCoLe* scenario when selecting only a subset of all possible pairs of classes within tasks. We vary the proportion of pairs kept and plot test accuracy.

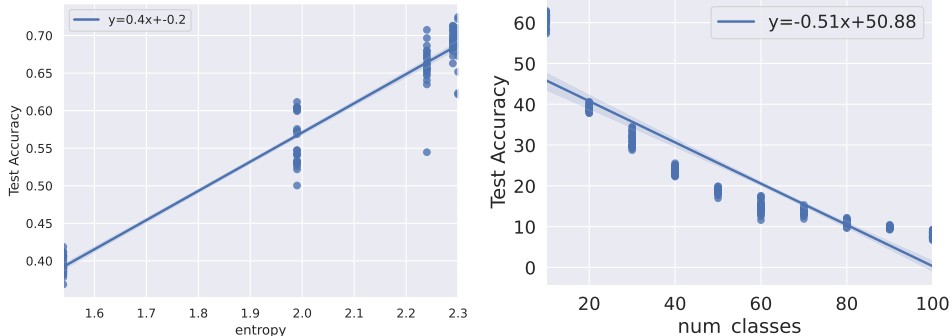

Figure 14: Regression of test accuracy over last 10 tasks on three seeds with entropy on CIFAR10 (left) and number of classes on CIFAR100 (right). More details of the experiments in Sec. 4

in class probabilities, we slightly modify this vector using $u' = u - \frac{1}{C} * numpy.arange(C) * \lambda$, with $C$ the number of classes and $\lambda$ and the hyperparameter that decides how much the distribution is modified. We choose empirically $\lambda = \frac{1}{2*C}$ for a slight imbalance. To increase the imbalance in the distribution, we multiply $u'$ by itself $d$ times. The complete experimentation tests the values of $0, 1, 2, 5$ and $10$. Note that $d = 0$ means that the distribution is uniform (cf. Appendix Sec. G.2 for the Python implementation and the probability vectors for each $d$).

Probability vectors for ten classes and $\lambda = 0.05$ with different $entropy\_decrease$ parameters (rounded with 3 decimals).

- $entropy\_decrease = 0 \rightarrow [0.1, 0.1, 0.1, 0.1, 0.1, 0.1, 0.1, 0.1, 0.1, 0.1]$
  entropy=2.303

- $entropy\_decrease = 1 \rightarrow [0.129, 0.123, 0.116, 0.11, 0.103, 0.097, 0.09, 0.084, 0.077, 0.071]$
  entropy=2.285

- $entropy\_decrease = 2 \rightarrow [0.161, 0.145, 0.13, 0.116, 0.103, 0.091, 0.079, 0.068, 0.058, 0.049]$
  entropy=2.237

- $entropy\_decrease = 5 \rightarrow [0.264, 0.204, 0.156, 0.117, 0.087, 0.063, 0.044, 0.031, 0.021, 0.013]$
  entropy=1.985

- $entropy\_decrease = 10 \rightarrow [0.424, 0.254, 0.148, 0.083, 0.046, 0.024, 0.012, 0.006, 0.003, 0.001]$
  entropy=1.537

```python
# num_classes: the total number of classes
# lambda: hyper-parameter = 1/(2*C)
# entropy_decrease: parameter that control the scale of the
                                        imbalance

import numpy as np
prob_vec = np.ones(num_classes) / num_classes
% introduction of slight imbalance
prob_vec = prob_vec - (1/num_classes) * np.arange(num_classes) *
                                        lambda
prob_vec /= prob_vec.sum()
prob_vec = prob_vec**entropy_decrease / (prob_vec**
                                        entropy_decrease).sum()

# we shuffle the vector so for each experiments the imbalance is
                                        not the same
np.random.seed(config.seed)
np.random.shuffle(prob_vec)

for task_indef in range(num_tasks):
    selected_classes = np.random.choice(np.arange(10), p=prob_vec
                                        , size=2, replace=False)
    # the we can create the task and train the model
    [...]
```

Figure 15: Pseudo-Code to control the distribution imbalance in classes.

## H  COMPUTE

This project was realized with the use of internal clusters. Each run was performed with a single GPU, mostly NVIDIA GeForce RTX 2070, Quadro RTX 8000, and Tesla V100-SXM2-32GB. The total amount of time required for the runs is 202 days.

## I  ARCHITECTURES NEURAL NETWORK FIRST EXPERIMENT

Architecture of the convolutional neural network used in Sec.3.

```python
import torch.nn as nn

relu = nn.ReLU()
conv1 = nn.Conv2d(1, 10, kernel_size=5)
conv2 = nn.Conv2d(10, 20, kernel_size=5)
maxpool2 = nn.MaxPool2d(kernel_size=2)
fc1 = nn.Linear(320, 50)
head = nn.Linear(50, 10)

# Forward pass with pytorch
# x dimension is [1,28,28]
x = relu(maxpool2(conv1(x)))
x = relu(maxpool2(conv2(x)))
x = relu(fc1(x))
x = head(x)
```

Figure 16: Pseudocode describing the architecture used for experiments in Sec.3

## J  FORGETTING

In long task sequences it becomes difficult and costly to track the forgetting on all tasks seen so far, hence we estimate forgetting by looking locally how learning a new task makes the model forget the one just before. We calculate local forgetting, which is the amount of forgetting in a task induced by learning the next task. Note: we only compute forgetting with non-overlapping classes between two tasks.

$$F_{local}(t) = \frac{1}{N-C} \sum_{j \notin \mathcal{Y}_t} A_{t,y=j} - A_{t-1,y=j} \tag{1}$$

With $\mathcal{Y}_t$ the set of classes in task $t$ and $A_{t,y=j}$ the accuracy realized on class $j$ at task $t$. $F_{local}(t)$ averages the forgetting generated in classes that are not in the current task. Then, total forgetting $F$ averages local forgetting on tasks seen to far and is computed as:

$$F = \frac{1}{T-1} \sum_{i=1}^{T} F_{local}(t) \tag{2}$$

## K  ACCUMULATION AND IID

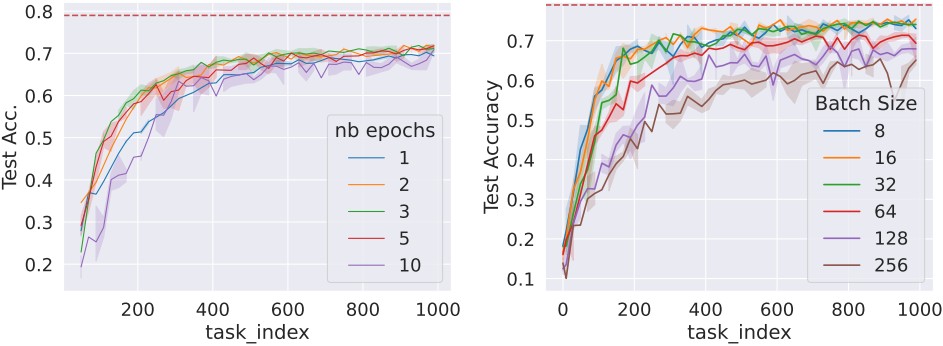

Figure 17: We simulate situation closer to IID training by changing the epochs number per task (top) (lower is closed to iid) [3] and growing batch size (bottom). Getting closer to the IID setting does not necessarily lead to better performance.

As discussed in Section 4.3, on a fixed number of epochs, increasing the batch size brings the training closer to IID training, while for a fixed batch size, growing the number of epochs per task makes the training further from IID. In this section, we investigate whether scenarios closer to IID training provide better performance. Growing the number of epochs or decreasing the batch size have the same impact: both increase the number of gradient steps within a task. There are, therefore, two forces that oppose each other, learning the current task and forgetting the past ones. On average, the algorithm should learn more about the current task than forget about the past ones to expect to be able to converge to a global solution.

**Setting:**  In this setting, we experiment with CIFAR10 with 2 classes per task. Then we vary (1) the batch sizes and (2) the number of epochs per task to evaluate if a minimum number of epochs with a maximum batch size (the closest to IID training) maximizes performance.

**Results:**  Fig. 17 (right) shows that the decrease in batch size during one epoch, i.e., getting further away from the IID training, increases the knowledge accumulation speed and final performance. We hypothesize that this result is due to a higher number of gradient steps for a smaller batch size. If that was the case, experiments with different batch sizes would lead to the same accuracy, but with a diverse number of tasks. However, in Fig. 17 (right), a larger batch size leads to a lower accuracy, which discards the hypothesis. Hence, in this experiment, moving further from IID training performs better, which is counterintuitive. An explanation for this is that, at the beginning of a task, knowledge

accumulation is superior to the knowledge degradation caused by forgetting. Therefore, training longer on each task improves the final knowledge accumulation. In the same way, Fig. 17 (left) shows that increasing the number of epochs per task accelerates knowledge accumulation in the number of tasks seen. However, if we grow the number of epochs to a certain point, the knowledge accumulation slows down. This figure shows that there is a trade-off to find in terms of number of gradient steps to converge faster and better.

To conclude, this experiment shows that training in settings that are as close as possible to IID (low number of gradient steps per task) does not always improve knowledge accumulation speed and performance.

## L    INFLUENCE OF EXACT DATA REPETITION

In most of our experimentation, when a class reappears, the exact same data is fed to the model. In this experiments, we want to investigate the influence of this. We compare training with exact same data and with data modified by random data augmentation.

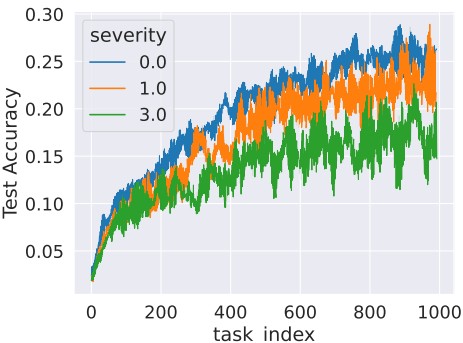

Figure 18: TinyImagenet 50/5, 1 epoch per task: Addition of random augmentation at each task with various severity. The augmentation slows down knowledge accumulation but does not prevent .

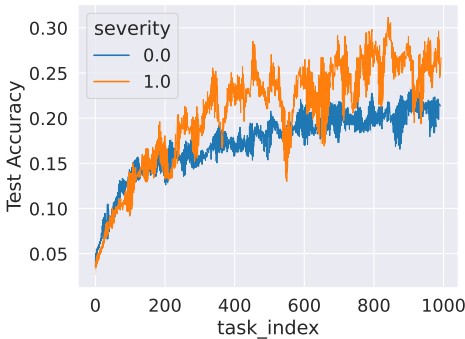

Figure 19: TinyImagenet 50/5, 5 epochs per task: when growing the number of epochs per task, the a slight data augmentation improves results of exact data repetition.

**Setting:** We use the augmentations proposed in Hendrycks & Dietterich (2019) that simulate common perturbation and corruption proposed in images. At each task, a new augmentation is selected and applied with severity 1. We selected our augmentation among, "no corruption", "gaussian noise", "shot noise", "impulse noise", "speckle noise", "gaussian blur", "defocus blur", "motion blur", "zoom blur", "fog", "snow", "spatter", "contrast", "brightness", "saturate", "elastic transform" and "glass blur". The minimized the chance of having the same exact data several times, and the augmentation applied to the data is variate and significant. We use a subset of 50 of the TinyImagenet dataset with 5 classes per task.

**Results:** Our results are presented in Fig. 20. It shows that the augmentation applied to each image to minimize the chances of having the exact same images does not compromise the accumulation of knowledge. Moreover, Fig. 19 shows that when growing the number of epochs per task, the augmentation can improve performance over exact data repetition.

## M  FREQUENCY OF OCCURRENCE VS CLASS ACCURACY

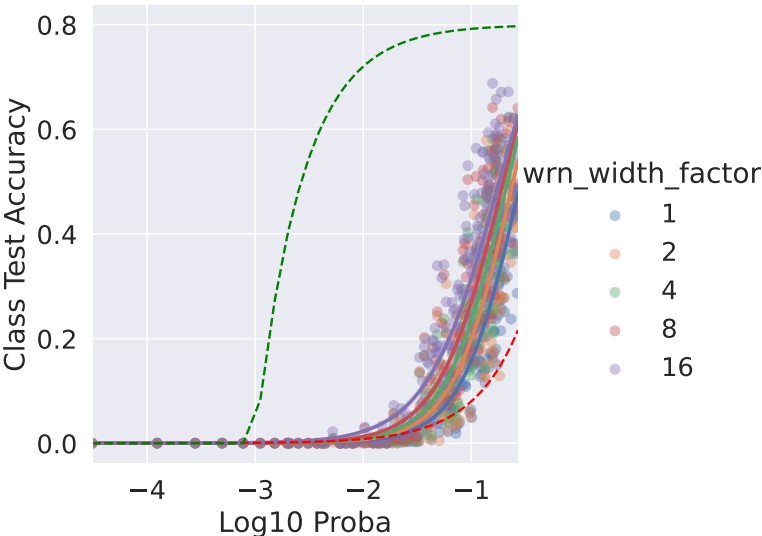

Figure 20: Lower bound - - - the model learn at 80% accuracy and forgets directly after, Upperbound - - - the model learnd up to 80% each task and never forget after. This representation helps to understand the continual learning capabilities of models at each frequency of re-occurence. Each point is the expected performance for a given probability given the model on three different seeds.

To understand better how model learn and forgets, we can analyze the expected performance for various frequency of occurrence.

**Setting:** Our goal is to better understand continual learning capabilities of models and algorithms by representing the expected performance on each class versus frequency of occurrence. In this setting, we train on CIFAR100 dataset with SGD ($lr = 0.1, momentum = 0$) on one epoch per task. All classes have a different probability to be sampled when building a task. We use wide resnet model to show that this representation helps to disentangle models capabilities.

**Results:**  The results of this experiments, show that we can differentiate models off different capabilities through this lens. This representation helps to describe capabilities of models or optimizer and better understand final performance. We also propose to plot the behaviour of a model that would catastrophically forget or a model that perfectly remember assuming a performance of 80% on each class when training on it.

