# OpenReview forum: "Challenging Common Assumptions about Catastrophic Forgetting"
_ICLR.cc/2023/Conference — Submitted to ICLR 2023_

### Official Review · Reviewer_3eiy · 2022-10-17

**Confidence:** 4
**Correctness:** 4
**Technical Novelty And Significance:** 1
**Empirical Novelty And Significance:** 1
**Recommendation:** 3

**Clarity, Quality, Novelty And Reproducibility:**

- Clarity: While the paper is organised in a somewhat non-standard manner, it is relatively easy to follow, reading more like a report of experiments carried out subsequently. Graphics are of limited quality however, requiring them to be read on a computer so one can zoom. Labels occasionally overlap with data (Figure 5 left) or are not centered or even out of the page margin (Figure 3 top). Figure 4 (c) has a double grid structure (in the background) and manually drawn black lines. Numbering is inconsistent (sometimes plots are labelled (a)-(z), sometimes there is no labelling). Gives an overall impression of a rushed piece of work with little attention to detail.
- Reproducibility: I have no concerns about reproducibility.
- Novelty: No new insights.

**Strength And Weaknesses:**

Weaknesses:
- The manuscript does not contain any new or unexpected insights. The fact that task re-occurrence lessens the problem of Continual Learning is not only well known but arguably the main reason that a large part of the CL literature is dedicated to rehearsal and pseudo-rehearsal techniques (i.e. the deliberate enforcing of re-occurrence, proposed as early as 1995). The authors seem to be aware of this "The reoccurence of data could be compared with the action of a replay methods" but do not carry out such a comparison. It is expected that a suitable replay technique would perform significantly better, especially in the more difficult SCoLe settings (e.g. Figure 5 left). It is also worth mentioning that scheduled task reoccurrence did indeed feature in highly cited earlier CL publications (e.g. [1][2]) but was regularly critiqued and does disappeared from the literature unless present in real CL problems (e.g. language modelling). Furthermore, the masking of the output layer is an existing technique well known and discussed in other works.
- The abstract does not clearly mention re-occurrence in the data distribution, merely "long sequences of tasks sampled from a finite environment" initially leading me to believe the authors found a profound insight that questioned the entire existence of the catastrophic forgetting phenomenon. This should be rephrased so as not to be misleading.
- SCoLe itself can barely be considered a contribution and would in most other submissions be merely considered the specific evaluation setting used in this work.

[1] Kirkpatrick, James, et al. "Overcoming catastrophic forgetting in neural networks." Proceedings of the national academy of sciences 114.13 (2017): 3521-3526.
[2] Schwarz, Jonathan, et al. "Progress & compress: A scalable framework for continual learning." International Conference on Machine Learning. PMLR, 2018.


**Summary Of The Paper:**

This paper proposes a new evaluation setting for Continual Learning in the setting of large number of tasks with recurring tasks. The authors propose SCoLe, an experimental framework for this setting and report their findings throughout the manuscript. The key claimed insight is the observation that Continual Learning is possible in a specific setting (reoccurring tasks, SGD without momentum, masking of output layer) without further restrictions.

**Summary Of The Review:**

This paper simply states a well-known observation and produces no new insights.

---

> ### Author Response · Authors · 2022-11-18
> **Answer to Reviewer 3eiy**
>
> Dear reviewer 3eiy,
>
> Thanks for your review. We try to answer it point by point in the following list. Also, we discuss more in-depth, more general concerns raised by several reviewers in the global answer.
>
> > The fact that task re-occurrence lessens the problem of Continual Learning is not only well known but arguably the main reason that a large part of the CL literature is dedicated to rehearsal and pseudo-rehearsal techniques
>
> Note that our work is not about a method that replays data to alleviate catastrophic forgetting. Our work is an empirical study about how different frequencies of data reoccurrence affect continual learning of deep learning models.
>
> On the other hand, replay aims at simulating IID settings. It does not simulate the reoccurrence of past tasks but merges several data distributions together.
>
> Indeed, rehearsal is often implemented as dense re-occurrence that simulates iid setting, not as a sparse re-occurrence, the question of how often the data should be repeated is often shortcutted by just mixing old and new in a fixed amount. From a broader point of view, our study could potentially help to optimize how replay is used. The fact that there is long-term remembering means that replay could be much more sparse than usually applied. Replay could be almost used only before evaluation (when it is possible).
>
> > scheduled task reoccurrence did indeed feature in highly cited earlier CL publications but was regularly critiqued and does disappeared from the literature unless present in real CL problems.
>
> Reoccurrence here is not scheduled. The critics of those publications usually target the fact that it makes the evaluation less clear. While our setups by varying the probability of reoccurrence, we can analyze results in a clearer and more rigorous way. Our goal is to evaluate gradient descent algorithms in continual learning setups in a new light.

---

### Official Review · Reviewer_DHkU · 2022-10-23

**Confidence:** 4
**Correctness:** 3
**Technical Novelty And Significance:** 3
**Empirical Novelty And Significance:** 3
**Recommendation:** 6

**Clarity, Quality, Novelty And Reproducibility:**


Overall, the paper is well-written, and the claims are supported by experimental results. While the authors do not provide code, they have explained the experimental setting clearly. In terms of novelty, the paper provides several interesting contributions regarding the knowledge accumulation in continual learning. However, I find breaking the no-revisiting assumption of continual learning too relaxing.


**Strength And Weaknesses:**


**Strengths:**

- I believe the SCoLe framework is interesting, and showing that DNNs are capable of knowledge accumulation for a long time is a novel contribution.

- The experiments are carefully designed and can support the claims of the paper.




**Weakness:**
- The main assumption of the SCoLe framework is not aligned with the typical CL assumption of not revisiting the past. In many real-world scenarios, the same data will not reoccur in the future. Hence, I believe it would be interesting if the authors could modify at least one experiment to validate their hypothesis on a more challenging benchmark. For instance, the authors could augment the images of each task differently and, by controlling the augmentation, measure the knowledge accumulation under various settings. I believe this experiment is crucial for the paper.


**Questions:**
- Could the authors explain the "gradient masking" mechanism more clearly? Isn't masking the gradients based on the classes similar to having a multi-head classifier (at training time)?

**Summary Of The Paper:**


The paper studies Continual Learning (CL) under a specific framework called SCoLe (Scaling Continual Learning). The major difference between SCoLe and CL is the sparse and controllable appearance of previous tasks and data when the number of tasks is very large. In addition, the paper proposes under the SCoLe framework, DNNs trained with gradient descent only accumulate knowledge without any supplementary CL mechanism.


The authors provide various designed experiments to study knowledge accumulation via different optimization methods and the impact of data shift. They show that continual fine-tuning yields knowledge accumulation under many scenarios (especially online learning scenarios) when previous data reoccur.

**Summary Of The Review:**

Overall, while I find some assumptions of the SCoLe framework not realistic, I believe the work provides an interesting contribution regarding knowledge accumulation in continual learning. The contributions could have been stronger if the authors had validated their hypothesis under a more realistic benchmark (please see the weakness section).

---

> ### Author Response · Authors · 2022-11-18
> **Answer to reviewer DHkU**
>
> Dear reviewer DHkU,
>
> Thanks for your review. We try to answer it point by point in the following list. Also, we discuss more in-depth, more general concerns raised by several reviewers in the global answer.
>
> > the same data will not reoccur in the future
>
> We respectfully disagree with this statement.  For example, a bot that is learning to play chess, will witness many times the exact same position or game, and even a robot evolving in a finite environment will probably receive many times the same observations (or at least so similar that they can be assumed to be the same). On the other hand, one point of CL is that data will reoccur later, at least at deployment. Remembering something that will never happen again. In our setup, we try to investigate if the model completely forgets between two recurrences.
>
> > the authors could augment the images of each task differently and, by controlling the augmentation, measure the knowledge accumulation under various settings. I believe this experiment is crucial for the paper.
>
> We added this setup in the appendix. We modified images with random perturbation and corruption inspired by “Benchmarking neural network robustness to common corruptions and perturbations” Hendrycks et al. We observe that the augmentation of data to avoid exact repetition does not compromise knowledge accumulation. We believe that careful augmentation could potentially improve test performance.
>
> > Could the authors explain the "gradient masking" mechanism more clearly? Isn't masking the gradients based on the classes similar to having a multi-head classifier (at training time)?
>
> With gradient masking, the inference is realized in the same way as the single head. However, the loss is computed by only taking into account the classes currently in the mini-batch. It is equivalent to masking out the gradients of classes not currently present in the mini-batch.
>
> > I find breaking the no-revisiting assumption of continual learning too relaxing.
>
> If CF was completed in time under T tasks, then re-occurrence should not impact performance, which is the point of experimenting with various sparsity of recurrence in this paper. Eve, if we could find a connection between sparse re-occurrence and application scenarios, this is not our first goal. We aim to evaluate forgetting in a more precise and meaningful way than what the usual setups do.
>
> > I find some assumptions of the SCoLe framework not realistic
>
> Agents interacting with the world often deal with data and situations reoccurring (games, a surveillance camera or a robot in an industrial factory). If usually, the continual learning setups impose no re-occurrence of data, it is because it makes it easier to estimate forgetting and disentangle it from other factors.
>
> > While the authors do not provide code
>
> The code is provided in the appendix through an anonymized link.

---

### Official Review · Reviewer_E4AE · 2022-10-24

**Confidence:** 4
**Correctness:** 2
**Technical Novelty And Significance:** 2
**Empirical Novelty And Significance:** 2
**Recommendation:** 5

**Clarity, Quality, Novelty And Reproducibility:**

The clarity of the work is generally good, and I expect the reported results to be relatively easily reproducible.

As described above, the main issue with this paper is in terms of quality / novelty. The paper claims to make a surprising empirical observation, but purely based on a simple thought experiment this observation can be predicted. Moreover, there exist prior work (e.g., Stojanov et al., 2019, CVPR) with results that allow analogous conclusions.

**Strength And Weaknesses:**

I found this paper an interesting read. I think considering continual learning setups where tasks or classes re-occur is a promising research direction (although in itself not novel, see some references below) that could indeed lead to important new insights. But I’m afraid I do not think the current paper provides substantial new insights.

The main claim of the paper is that it challenges “common assumptions about catastrophic forgetting”. The premise for this claim is that a common assumption is that catastrophic forgetting equals forgetting everything and that therefore knowledge accumulation is not possible. The main result of the paper is experimentally disproving this “common assumption” by demonstrating that catastrophic forgetting does not necessarily equate to forgetting everything and that knowledge accumulation is possible.

It surprises me that the authors consider it to be a common assumption that catastrophic forgetting equals forgetting everything. I do not agree with this premise. Let’s consider a simple example. This assumption means that after training on a new task, information of past tasks is no longer contained in the network. In other words, a network first trained on task 1 and then on task 2, should have the same task 1 performance as a network only trained on task 2. It is clear that in general this is not the case. (Except perhaps for class incremental settings, but this can be remedied with a simple trick: see for example https://arxiv.org/abs/2106.01834, which the authors are aware of.)

With this premise gone, the paper loses its fundament and main contribution.

In Fig 3 the authors make an attempt to convince the reader that the assumption that catastrophic forgetting corresponds to forgetting everything is common / reasonable, but this example is problematic (in fact, it might even disprove the point the authors try to make).
In Fig 3A, it is shown that when the authors train a standard neural network in a standard way on their protocol, there is no increase in performance over time. The authors say this result is in accordance with “the common catastrophic forgetting phenomenon”. But it actually is not. If always everything would be forgotten except for the current task, performance should be constant near 20%. Performance however drops to 10%. I expect this might be due to the learning rate being too high and the model collapsing onto one particular class.

Finally, I’d like to encourage the authors to take note of the following papers that also consider continual learning setups where tasks or classes re-occur:
- Stojanov et al. (2019, CVPR) https://openaccess.thecvf.com/content_CVPR_2019/html/Stojanov_Incremental_Object_Learning_From_Contiguous_Views_CVPR_2019_paper.html
- Cossu et al. (2022, Frontiers AI) https://www.frontiersin.org/articles/10.3389/frai.2022.829842/full

Especially the first paper seems to have results that allow analogous conclusions to those presented in the current paper.


**Summary Of The Paper:**

The main claim of the paper is that it challenges “common assumptions about catastrophic forgetting”. The premise for this claim is that a common assumption is that catastrophic forgetting equals forgetting everything and that therefore knowledge accumulation is not possible. The main result of the paper is experimentally disproving this “common assumption” by demonstrating that catastrophic forgetting does not necessarily equate to forgetting everything and that knowledge accumulation is possible.

**Summary Of The Review:**

I found this paper an interesting read. I think considering continual learning setups where tasks or classes re-occur is a promising research direction (although in itself not novel, see some references below) that could indeed lead to important new insights. But I’m afraid I do not think the current paper provides substantial new insights.

---

> ### Author Response · Authors · 2022-11-18
> **Answer to reviewer E4AE**
>
> Dear reviewer E4AE,
>
> Thanks for your review. We try to answer it point by point in the following list. Also, we discuss more in-depth, more general concerns raised by several reviewers in the global answer.
>
> > a common assumption is that catastrophic forgetting equals forgetting everything and that, therefore knowledge accumulation is not possible. The main result of the paper is experimentally disproving this “common assumption” by demonstrating that catastrophic forgetting does not necessarily equate to forgetting everything and that knowledge accumulation is possible. It surprises me that the authors consider it to be a common assumption that catastrophic forgetting equals forgetting everything.
>
> We explain in the global answer why we claim that catastrophic forgetting being complete is a common assumption.
>
> In addition, The point of our work is not only to disprove this “common assumption” but to analyze the accumulation of knowledge more closely. In the beginning, indeed, we show that catastrophic forgetting is not complete, but in the following experiments, we investigate many factors of variation: number of epochs, batch size, number of classes per task, the total number of classes, long-term shifts, the entropy of class distribution...
>
> > If always everything would be forgotten except for the current task, performance should be constant near 20%.
>
> Thanks for pointing this out. We revised this figure by tuning the learning rate. While it does not change our conclusion, it was indeed collapsing for only one class, but this class was changing from one task to another. Is there literature on this subject?
>
> > Except perhaps for class incremental settings, but this can be remedied with a simple trick
>
> Masking solves the problem in the classification layer, however it does not solve the problem completely. In TIL, when the model does not forget completely it is not clear if it is because of feature reuse or because of low forgetting.
>
> > I’d like to encourage the authors to take note of the following papers that also consider continual learning setups where tasks or classes re-occur
>
> Thank you! We will incorporate them in the related work.

---

> > ### Comment · Reviewer_E4AE · 2022-11-19
> > **Some quick comments / questions**
> >
> > Thanks to the authors for the rebuttal. In the hope to still give the authors the opportunity to reply, I already post a few quick comments / questions.
> >
> > Firstly, I appreciate the explanation of the authors regarding their initial experiments. However, fixing these experiments by post hoc tuning the learning rate is problematic, as these new experiments clearly are not "initial experiments" anymore, while they are still labeled as such.
> >
> >  Secondly, I appreciate the comment of the authors that they will incorporate the related papers I highlighted into their "related work" section. However, as far as I can see this has not been done yet. When are the authors planning to do this?

---

> > > ### Author Response · Authors · 2022-11-19
> > > **Answer**
> > >
> > > Thanks for those quick comments/questions,
> > >
> > > Sorry for not putting the related papers. This is an unfortunate mistake, we will add them in the final version if the paper gets accepted.
> > >
> > > Concerning the "Initial experiments", they are still "initial" in our opinion since they are the starting point of our study and the presentation of the initial problem/observation. Tuning the learning rate in those experiments improved the results while maintaining the problem, hence in our opinion, it stays a valid representation of what usually happens in continual learning experiments.

---

> > > > ### Comment · Reviewer_E4AE · 2022-11-21
> > > > **Thanks for quick response**
> > > >
> > > > Thanks for the quick response to my comments.
> > > >
> > > > I think that is a rather generous interpretation of "initial experiments", and my worry is that many readers might not interpret this as such. I would suggest the authors to rephrase the description of these experiments.
> > > >
> > > > Regarding the discussion of the "related papers", my interest is not so much in whether these papers will be cited, but more in how they will be discussed. In particular I'd be interested in what the authors think the new contributions/insights provided by their paper are taking into consideration the paper by Stojanov et al. (2019). I would be happy to still take this into account if the authors post their discussion of these related works here.

---

> > > > > ### Author Response · Authors · 2022-11-21
> > > > > **Answer**
> > > > >
> > > > > This is a legitimate request.
> > > > >
> > > > > There are two points for Stojanov et al. (2019):
> > > > >
> > > > > 1 - The authors investigate the repetition of classes and instances and show that it reduces the effects of catastrophic forgetting on continual learning algorithms. -> In our experiments, the repetition of tasks/classes is achieved to better understand how DNNs forget without continual learning algorithms.
> > > > >
> > > > > 2 - Stojanov et al (2019) also proposed the CRIB environment to simulate infant learning exposure, making it possible to generate an unlimited data stream. On the other hand, Cossu et al. propose a scenario with repetition and argue it is more realistic to experiment without "the no repetition constraint" of usual CL scenarios and ease the problem.
> > > > > -> In our experiments, the scenarios are generated, such as varying the frequency of occurrences and the divergence to IID and putting it in perspective with forgetting.

---

### Official Review · Reviewer_Pyx6 · 2022-10-24

**Confidence:** 5
**Correctness:** 2
**Technical Novelty And Significance:** 1
**Empirical Novelty And Significance:** 1
**Recommendation:** 3

**Clarity, Quality, Novelty And Reproducibility:**

While the writing is clear, the paper lacks the novelty. Please refer to the strengths and weaknesses section above for details.

**Strength And Weaknesses:**

**Strengths**

1. The premise of studying continual learning in the presence of large number of tasks and drawing conclusions from it is a worthwhile goal.
2. Overall the paper is well-written and easy to follow.

**Weaknesses**

1. **Contrived setup**: Overall the setup seemed very contrived and does not justify the claims regarding catastrophic forgetting made in the paper. The authors took small datasets MNIST, CIFAR10 & 100 etc, and repeatedly sampled large number of tasks (up to 1000) consisting of small number of disjoint classes, sometimes even 1 class per task, repeating classes between the tasks, and measured the trajectory of test performance of that dataset (on all classes). It is quite intuitive to think that in such a setup if you control the number of epochs per task properly and keep the total number of classes low, you’ll closely approximate the IID training and your test set performance would increase overtime (barring overfitting). But this doesn’t tell you anything about the catastrophic forgetting. It is quite intuitive that if you keep on observing new classes (along with repeatedly observing previous classes) your overall test performance would keep on increasing.

2. **When the setup is closer to IID training there is less catastrophic forgetting, when the setup is far there is more CF**: Even in the given setup, if one closely looks at the experiments, it seems that when one is closer to the IID setting, the overall test set performance remains high. Similarly, if one moves away from the IID setting, the test set performance degrades. For example, in Fig 7 when the authors compare single-epoch (closer to IID) vs multi-epoch training (farther from IID), and experiments are slightly challenging CIFAR100 (N=50) there is a significant gap between the single- vs multi-epoch performance. It seems that only in very specific situations, when the total number of classes are small and the SGD steps per task are small, or when the setting is closer to IID, the test set performance increases with the number of tasks. This makes the overall results less interesting and trivial.


3. **Measuring CF through test set performance**: The notion of forgetting is not clear in this work. What the authors measure is the full test set performance and show that with the data/ task repetition that performance increases. If the data is removed from the mix, cf. Fig 6, the performance of those classes degrades. All this one can observe in a single-task multi-epoch training as well. Perhaps the authors could properly define the notion of forgetting. For example, measuring the performance of a task when it is seen again and plotting that over time.

4. Overall, the learning setting seems contrived and experiments do not justify the claims made in the paper.  Some results on increasing the width resulting in less forgetting (Mirzadeh et al.), and warm starting decreasing the generalization performance (Ash & Adams) are already well-known in the literature.


**Summary Of The Paper:**

The paper argues that when the number of tasks are increased in a continual learning setting, catastrophic forgetting ceases to be a problem and a simple SGD training with gradient masking keeps on accumulating knowledge. Towards this, the authors propose a framework, called SCoLe, where the authors repetitively sample tasks from a fixed number of classes of small datasets. The authors show that the full test set performance increases with the number of tasks, and that if a training setting moves away from the IID training, for example by increasing the number of epochs per task or the total number of classes in a dataset, the test set performance remains low. Overall, the proposed claims are not well-justified and the training setting looks trivial – closer to IID training.


**Summary Of The Review:**

Please refer to the strengths and weaknesses section above for details.

---

> ### Author Response · Authors · 2022-11-18
> **Answer to reviewer Pyx6**
>
> Dear reviewer Pyx6,
>
> Thanks for your review. We try to answer it point by point in the following list. Also, we discuss more in-depth, more general concerns raised by several reviewers in the global answer.
>
> > The paper argues that when the number of tasks is increased in a continual learning setting, catastrophic forgetting ceases to be a problem
>
> We respectfully disagree, the main point of the paper is to empirically study in depth how a model learns and forgets through long sequences of tasks.
>
> > by increasing the number of epochs per task or the total number of classes in a dataset, the test set performance remains low.
>
> By increasing the number of classes, the training becomes longer, but the accuracy keeps increasing. This highlights that knowledge accumulation outweighs forgetting to the extent that is dependent on the level of non-stationarity in the stream.
>
> Additionally, increasing the number of epochs might keep the test set performance low, but it is only sometimes the case (ex: Fig 7 left, and appendix K fig 17 left).
>
> *> It is quite intuitive to think that in such a setup if you control the number of epochs per task properly and keep the total number of classes low, you’ll closely approximate the IID training and your test set performance would increase overtime (barring overfitting).*
>
> Indeed keeping the number of classes per task high and the number of epochs per task low would make the setup closer (not equivalent) to IID training (see appendix B for formal statement). In our experiments, however, we control this by varying the parameters mentioned to ensure our setting is not equivalent to IID. Hence, in the setup that is the closest to IID the total number of classes is 10, and classes per task is 2. With one epoch per task, it means classes appears once every 5 epochs on average (which represents around 156 update steps with cifar10 and batch size of 64 and, therefore, not an approximation of IID).
>
> Given current knowledge about catastrophic forgetting occurring already within a single epoch in CL (**[3], Fig 3 )** we expect catastrophic forgetting to outweigh the knowledge accumulation and keep the overall test performance low. In contrast, we observe that the overall test performance keeps increasing even for high levels of non-stationarity (CIFAR 100 with 2 classes per task). We interpret it as evidence for knowledge accumulation outweighing catastrophic forgetting.
>
> [3]  “REMIND Your Neural Network to Prevent Catastrophic Forgetting” Hayes et al ECCV 2020
>
> >It is quite intuitive that if you keep on observing new classes (along with repeatedly observing previous classes) your overall test performance would keep on increasing.
>
> We argue that given the current knowledge on catastrophic forgetting, overall performance on test set could stay very low, grow until some upper bound significantly lower than IID baseline or reach similar performance as IID. Our setup has not been experimented on before.
>
> *> Training with class reoccurrence doesn’t tell you anything about the catastrophic forgetting.*
>
> Note that catastrophic forgetting is not a discrete event. Since we control the frequency of recurrence of tasks and classes, we can observe how fast the model forgets. If forgetting was complete before recurrence, then there would be no progress through time. The test set accuracy provides insight into the learning vs forgetting trade-off.
>
> *> When the setup is closer to IID training, there is less catastrophic forgetting, when the setup is far there is more CF*
>
> This is only sometimes true, multiple epochs may improve performance over a single epoch, as in fig 17 left (multiple epochs being further from iid than a single epoch).
>
> Learning and forgetting are opposed to each other, leading to globally improving or degrading overall performance. While usually learning longer can lead to less overall improvement, learning can be stronger than forgetting several epochs.
>
> > *Measuring CF through test set performance. The notion of forgetting is not clear in this work. What the authors measure is the full test set performance and show that with the data/ task repetition that performance increases.*
>
> Forgetting is typically defined as the drop in performance in past tasks. Since data and tasks might reoccur and the sequence of past tasks might be very long, we estimate it by measuring how the accuracy decreases on all classes not currently available ( Fig 3.d). In most of the paper, our goal is to estimate if the model learns more than it forgets. Hence measuring this test accuracy of the full test gives a good estimate of progress. If forgetting is stronger than learning, the overall test performance will decrease or stagnate.

---

> > ### Author Response · Authors · 2022-11-18
> > **end of answer**
> >
> >
> > > Overall, the learning setting seems contrived, and experiments do not justify the claims made in the paper
> >
> > What we claim is that the term catastrophic forgetting is maybe too alarmist since the model does not forget completely. In usual setups, catastrophic forgetting might appear to indeed forget catastrophically and thus, we design a setup that shows that remembering can be stronger than forgetting in the long term and that helps to analyze this forgetting.

---

> > > ### Comment · Reviewer_Pyx6 · 2022-11-22
> > > **Rebuttal acknowledgment**
> > >
> > > I read the authors' rebuttal. Unfortunately, neither of the concerns were properly addressed. The authors' relied on a lot of hand wavy arguments like, given our knowledge of _this_ _that_ might have happened, which, in my opinion, is not a good science. In other instances, the authors rebuttal relied on 'yes this is the problem but only sometimes' (cf. their response to point 1 and 2 in the main concerns). Which makes it difficult to glean a unified message from the paper.
> > >
> > > I could see that the authors have put some effort in writing the rebuttal. I acknowledge that. But, unfortunately, in its current form the paper is not ready for a full publication.

---

### Author Response · Authors · 2022-11-18
**Global Answer**

We would like to thank our reviewers for their answers,

We summarize here some of the important points mentioned in the reviews.

First, we thank the reviewer for highlighting the strength of the paper “interesting setting” (E4AE, DHkU), good experimentation (DHkU), well written/easy to follow/clear (E4AE, Pyx6, 3eiy), good reproducibility (E4AE, 3eiy).

We address here some concerns raised.

- The common assumption that catastrophic forgetting makes the model forget completely/in depth.

In our point of view, the way deep neural networks forget is not yet understood well. On one hand side, the fact that forgetting is usually addressed with the adjective “catastrophic”, and not “gradual” or “relative”, literally suggests there is a common assumption that loss in performance in CL is directly linked with the permanent and drastic erasure of information.

Empirically this can be seen, for example, in class incremental learning where performance in past classes decreases quickly to zero. ([2] Fig 10)

Even in generative modelling, it has been shown that GANS and VAEs completely forget to generate samples from previous distributions when trained on new ones e.g. see Fig. 14 in [1].

Usually, when designing algorithms for CL the emphasis lies on preventing this presumably “catastrophic” forgetting.  These algorithms are designed under the assumption of forgetting being “catastrophic” (or complete) even though this might not be true (e.g. EWC, IMM, DR, EWC_KFAC, OGD, Icarl, PodNet, UCIR, BiC,… etc.).

- contribution & novelty

One of the potential reasons for the assumption of forgetting being “catastrophic” lies, in our opinion, in the fact that the classical CL setting does not allow to observe knowledge accumulation with standard SGD-based training. The aim of this paper is to point out that knowledge accumulation may indeed overweight forgetting in long sequences of tasks in a closed-world scenario (i.e. reoccurring concepts), to provide some tools to analyze it and show some experiments to investigate the influence of change in non-stationarity. More generally, beyond permut MNIST scenarios, our work is the first one to propose a scenario where the number of tasks can scale and to study continual learning phenomena in such a scenario.

From a practical point of view, in the classical CL scenarios, it is not taken into account that the non-stationarity level may vary (so it's not just iid vs. non-iid) — i.e. arguably, the learning environment of any biological learning system features reoccurrence of concepts and hence is not completely non-stationary. Hence, we think it is important to take this into consideration when designing new algorithms. One straightforward example is when designing replay algorithms, there is, in fact, no need to simulate a completely iid distribution if the goal is to accumulate knowledge. This could be of particular importance when training on large datasets incrementally, and it may be computationally infeasible to simulate completely iid settings through replay.

[1] "Generative models from the perspective of continual learning." Lesort et al.,  *2019 International Joint Conference on Neural Networks (IJCNN)*. IEEE, 2019.

[2] “A Neural Dirichlet Process Mixture Model for Task-Free Continual Learning” Soochan Lee et al. , ICLR 2020

We thank the reviewers again for their reviews, we added the experiments asked for and fixed the experimental bugs in some of the early experiments.

---

### Decision · Program_Chairs · 2023-01-20

**Decision:**

Reject

**Justification For Why Not Higher Score:**

The main contribution of this paper is a tool called SCoLe that can be used for studying the behavior of DNN in continual learning of arbitrary long task sequences.
The reviewers shared the concerns: the main assumption of the SCoLe framework is not as challenging as the typical CL one of no data/task re-occurrence; similar CL setting has been studied before, and the paper does not provide new insights.
The authors' rebuttal failed to address the above concerns.
Although the setting is interesting and could lead to new insights of CL, I think the paper is not ready for publication in its current form due to the weaknesses listed in the summary.

**Justification For Why Not Lower Score:**

N/A

**Metareview: Summary, Strengths And Weaknesses:**

This paper investigates the behavior of DNN in continual learning of long task sequences. The authors first develop a tool called SCoLe to generate CL setting, then employ the tool to conduct CL experiments with SGD. They discover a phenomena that's not presented in CL of short task sequence -- catastrophic forgetting does not happen and knowledge accumulates over time when classes and tasks repeat in the future and gradient of unseen classes are masked.

Strengths:
- The authors developed a tool SCoLe that can be used for studying the behavior of DNN in continual learning of arbitrary long task sequences.
- They present a new phenomena not shown in shorter ones.
- It's well-written.

Weaknesses:
- The main assumption of the SCoLe framework is not as challenging as the typical CL one of no data/task re-occurrence. In many real-world scenarios, the same data will not reoccur in the future.
- The discovered phenomena does not necessarily happen in real CL scenario.
- Similar CL setting has been studied before:
Stojanov et al. (2019, CVPR)
https://openaccess.thecvf.com/content_CVPR_2019/html/Stojanov_Incremental_Object_Learning_From_Contiguous_Views_CVPR_2019_paper.html
Cossu et al. (2022, Frontiers AI) https://www.frontiersin.org/articles/10.3389/frai.2022.829842/full
- The paper does not provide new insights.